# *Mycobacterium tuberculosis* FadD18 Promotes Proinflammatory Cytokine Secretion to Inhibit the Intracellular Survival of *Bacillus Calmette–Guérin*

**DOI:** 10.3390/cells13121019

**Published:** 2024-06-11

**Authors:** Yongchong Peng, Tian Tang, Qianqian Li, Shiying Zhou, Qin Sun, Xinjun Zhou, Yifan Zhu, Chao Wang, Luiz E. Bermudez, Han Liu, Huanchun Chen, Aizhen Guo, Yingyu Chen

**Affiliations:** 1National Key Laboratory of Agricultural Microbiology, College of Veterinary Medicine, Huazhong Agricultural University, Wuhan 430070, China; ycpeng@mail.hzau.edu.cn (Y.P.); conky@webmail.hzau.edu.cn (T.T.);; 2National Animal Tuberculosis Para-Reference Laboratory (Wuhan) of Ministry of Agriculture and Rural Affairs, International Research Center for Animal Disease, Ministry of Science and Technology, Huazhong Agricultural University, Wuhan 430070, China; 3Department of Biomedical Sciences, College of Veterinary Medicine, Oregon State University, Corvallis, OR 97331, USA

**Keywords:** *Mycobacterium tuberculosis*, FadD18, adhesin, invasion, inflammation, intracellular survival

## Abstract

*Mycobacterium tuberculosis* causes 6.4 million cases of tuberculosis and claims 1.6 million lives annually. Mycobacterial adhesion, invasion of host cells, and subsequent intracellular survival are crucial for the infection and dissemination process, yet the cellular mechanisms underlying these phenomena remain poorly understood. This study created a *Bacillus Calmette–Guérin* (BCG) transposon library using a MycomarT7 phage carrying a Himar1 Mariner transposon to identify genes related to mycobacteria adhesion and invasion. Using adhesion and invasion model screening, we found that the mutant strain B2909 lacked adhesion and invasion abilities because of an inactive *fadD18* gene, which encodes a fatty-acyl CoA ligase, although the specific function of this gene remains unclear. To investigate the role of FadD18, we constructed a complementary strain and observed that *fadD18* expression enhanced the colony size and promoted the formation of a stronger cord-like structure; FadD18 expression also inhibited BCG growth and reduced BCG intracellular survival in macrophages. Furthermore, FadD18 expression elevated levels of the proinflammatory cytokines IL-6, IL-1β, and TNF-α in infected macrophages by stimulating the NF-κB and MAPK signaling pathways. Overall, the FadD18 plays a key role in the adhesion and invasion abilities of mycobacteria while modulating the intracellular survival of BCG by influencing the production of proinflammatory cytokines.

## 1. Introduction

Before the onset of the coronavirus pandemic in 2019, tuberculosis (TB) was the primary cause of mortality [1]. The World Health Organization initiated “the End TB Strategy” with the aim of diminishing TB-related deaths and incidence rates by 95% and 90%, respectively, by the year 2035 in comparison to the 2015 levels; however, the worldwide burden of TB persists. In 2022, 7.5 million TB cases were reported with 1.67 million deaths, encompassing both HIV-negative and -positive individuals worldwide [1]. Enhancing the development of novel diagnostics and pharmaceuticals, as well as advancing TB treatments and preventive measures, are paramount global objectives outlined in the End TB Strategy.

*Mycobacterium tuberculosis* (*M. tb*) is the primary cause of TB in humans, has a robust cell envelope, and synthesizes a diverse array of lipids, including vital short-chain fatty acids and intricate mycolic acids [2]. These mycolic acids serve as the principal lipid constituents of mycobacterial cell walls, playing a crucial role in the structural integrity of the cell envelope, as well as contributing to virulence, pathogenicity, immune response, and resistance to antibiotics [3].

*M. tb* harbors 36 *fadD* genes, which can be classified into two distinct subclasses: fatty acyl-CoA ligases (FACLs) involved in lipid degradation and fatty acyl-AMP ligases (FAALs) primarily associated with lipid biosynthesis [2]. Conversely, *Escherichia coli* has only one *fadD* gene involved in fatty acid degradation, underscoring the significance of lipid metabolism in mycobacteria [4]. Previous studies have annotated and reported the functions of many of the proteins encoded by these *fadD* genes. For example, FadD5 recycles mycolic acids for the long-term survival of the mycobacterium [5]; the FadD8 enzyme plays a crucial role in catalyzing the binding of fatty acids containing a maximum of 18 carbon atoms to CoA, a fundamental step in the utilization and metabolism of cholesterol in mycobacteria [3,6]; FadD10 plays a role in the biosynthesis of a virulence-associated lipopeptide by facilitating the transfer of fatty acids to an acyl carrier protein [6]; FadD26 is essential for the generation of phthiodiolone dimycocerosates (DIMs); and FadD22 and FadD29 play crucial roles in a specific step of the PGL biosynthetic pathway, with FadD22 exhibiting p-hydroxybenzoyl-AMP ligase activity and FadD29 demonstrating FAAL activity [7].

In this study, a BCG transposon library was constructed and a mutant, designated as B2909, was characterized as lacking the functional *fadD18* gene. A comparative analysis with the wild-type BCG strain revealed the diminished adhesion and invasion capabilities of the mutant. Subsequent investigations were conducted to clarify the impact of FadD18 on *M. tb* infection in macrophages. Our results suggest that FadD18 inhibits the extracellular proliferation and intracellular persistence of BCG, while also promoting the production of proinflammatory cytokines through the NF-κB/MAPK-signaling pathways.

## 2. Materials and Methods

### 2.1. Bacteria and Cell Culture

The BCG Pasteur strain was graciously supplied by Luiz E. Bermudez of Oregon State University in the United States. Mycobacterial strains were cultivated in Middlebrook 7H9 (BD PharMingen, San Diego, CA, USA) broth supplemented with 10% oleic acid, albumin, dextrose, catalase (OADC; BD PharMingen, 0.5% (*v*/*v*) glycerol (Sigma-Aldrich, St. Louis, MO, USA), and 0.05% (*v*/*v*) Tween 80 (Sigma-Aldrich) or on Middlebrook 7H11 agar plates (BD PharMingen) containing 10% OADC.

A549 cells were cultured in Dulbecco’s Modified Eagle Medium (DMEM, HyClone, Logan, UT, USA) supplemented with 10% fetal bovine serum (FBS, Gibco, Grand Island, NY, USA) and seeded at a density of 10^5^ cells per well in a 12-well plate. THP-1 cells were cultured in RPMI-1640 (HyClone) medium supplemented with 10% FBS, stimulated with phorbol 12-myristate 13-acetate (PMA, Sigma-Aldrich) for 24 h to induce maturation, and utilized upon reaching 90% confluency for infection studies, with subsequent monitoring of cell viability.

### 2.2. Invasion and Adhesion Deficiency of BCG Mutant Selected via a BCG Transposon Library

The transposon library of BCG was created through Himar1 mutagenesis, resulting in a total of 3034 individual transposon mutants [8]. In the invasion and adhesion assay, A549 cells were exposed to wild-type or mutant BCG strains with a multiplicity of infection (MOI) of 10 bacteria to 1 host cell for 1 h at 37 °C for invasion or 30 min at 4 °C for adhesion. Subsequently, the remaining cells were washed three times with 1× phosphate-buffered saline (PBS, Gibco) and lysed with 0.025% Tween-20 (Sigma-Aldrich) for 10 min. A 10-fold serial dilution of the cell lysates was then performed before plating them onto 7H11 agar plates. After three weeks, the colony-forming units (CFUs) of bacteria were quantified.

### 2.3. Identification of the Transposon Insertion Site

Transposon mutant genomic DNA was extracted from the selected mutants as previously described and sent for sequencing based on the amplification on the transposon–transposon–chromosomal junctions [8]. In order to confirm the sequence outcomes, PCR was conducted using upstream primers originating from transposon sequences and downstream primers derived from sequences situated 500–700 base pairs downstream of the insertion site. Mutant genomic DNA was utilized as the template, with wild-type BCG genomic DNA serving as the negative control in the PCR protocol. The protocol included an initial denaturation step at 95 °C for 5 min, followed by 30 cycles of denaturation at 95 °C for 30 s, annealing at 60 °C for 1 min, and extension at 72 °C for 1 min, concluding with a final extension step at 72 °C for 5 min. Specifically, 5ʹ-CCTCGTGCTTTACGGTATCGC-3ʹ (forward) and 5ʹ-CGGTGCGATCAACACGAACGAC-3ʹ (reverse) primers were used for the PCR amplification.

### 2.4. Identification and Sequence Analysis of FadD18 In Silico

Bioinformatics tools were utilized for in silico identification and sequence analysis of FadD18. ProtParam was utilized for the evaluation of the aliphatic index and grand average of hydropathicity index (GRAVY) value [9], while the TMHMM web server was employed for the prediction of the transmembrane structure [10]. Homologous sequence alignment was carried out using the Clustal Omega and ESPript web server [11,12]. Furthermore, the NCBI CDD search function was utilized for the analysis of conserved domains within FadD family members, including FadD18 and 35 additional members [13]. The analysis of conserved motifs within the protein sequences was performed utilizing the MEME suite’s motif search tool [14].

### 2.5. Construction of BCG Complementary Strain

The CDS of *fadD18* gene (Gene ID: 888277) was amplified from BCG genomic DNA using specific primers (forward: 5ʹ-CGCGGATCCATGGCGGCATCTTTAAGTGAG-3ʹ and reverse 5ʹ-CCGGAATTCTCAGGAACCGCTCGTCACG-3ʹ). The amplified gene was then cloned into the pMV261 vector and transfected into BCG cells through electroporation to generate the complementary strain B2909C, which was validated by PCR using the forward primer.

### 2.6. Growth Curve and BCG Morphology

Growth curves were plotted based on the growth of BCG strains every 3 days at 37 °C in 7H9 liquid medium. A stereomicroscope (KEYENCE, version VHX-7000, Osaka, Japan) was used to observe and record the morphology of a single colony. A three-point rounding method was used to calculate the area of each individual colony. Five to ten individual colonies were measured per plate.

### 2.7. Intracellular Survival

An MOI of 10:1 was utilized to infect PMA-differentiated THP-1 cells with BCG strains. Subsequently, the cells underwent three washes with PBS and were incubated in a fresh medium containing 100 μg/mL gentamicin to remove extracellular bacteria. Following this, the cells were washed again and lysed with 0.025% Tween-20 for 10 min 0, 24, 48, and 72 h postinfection (PI). The lysed cells were then plated on 7H11 agar plates at a ten-fold dilution, and the colonies were enumerated after an incubation period of 3–4 weeks at 37 °C.

### 2.8. Real-Time PCR and Enzyme-Linked Immunosorbent Assay (ELISA)

An MOI of 10:1 was utilized to infect PMA-differentiated THP-1 cells with BCG strains, followed by extraction of total RNA and conversion into cDNA using HiScript II Q RT SuperMix (Vazyme, Nanjing, China). Subsequent qPCR amplification was carried out using a Bio-Rad IQ5 (Hercules, CA, USA) instrument with the following parameters: an initial denaturation step at 95 °C for 5 min, followed by 40 cycles of denaturation at 95 °C for 10 s and annealing/extension at 60 °C for 30 s. The primer sequences employed in this study can be found in Table 1. The β-actin gene was employed as an internal control for standardization purposes. The levels of TNF-α, IL-6, and IL-1β proteins in cell culture supernatants were quantified utilizing a commercially available ELISA kit (Neobioscience, Shenzhen, China) in accordance with the manufacturer’s guidelines.

### 2.9. Statistical Analysis

The assays and experiments were carried out in triplicate, with the data presented as means ± standard errors of the mean (SEMs) from the triplicates. Statistical analysis was conducted using GraphPad Prism software (Version 9.0, Boston, MA, USA), with Student’s *t*-test utilized for comparisons between two groups and analysis of variance (ANOVA) for comparisons among more than two groups. Statistical significance is indicated at four levels: non-significant (ns) *p* > 0.05, * *p* < 0.05, ** *p* < 0.01, and *** *p* < 0.001.

## 3. Results

### 3.1. Mutant B2909 Reduced the Invasion and Adhesion Abilities of BCG

The BCG transposon mutant library was created using the MycomarT7 phage containing the Himar1 Mariner transposon, generating 3034 mutant strains (Figure 1a,b). A screening process was conducted on >200 mutants to assess their invasion ability (Figure 1b). A further examination indicated that the mutant strain B2909 displayed reduced invasion and adhesion capacities in comparison to the wild-type BCG strain (Figure 1c,d). Transposon sequencing was employed to identify the specific gene that was inserted and inactivated in B2909, with the PCR validation confirming that the inactivated gene was *fadD18* (Appendix A).

### 3.2. Bioinformatics Analysis of fadD18

An in silico analysis was utilized to investigate the potential functional role of the FadD18 protein. The FadD18 protein displayed a notable abundance of hydrophilic amino acids, as evidenced by an aliphatic index of 67.11 and a GRAVY value of − 0.589, indicating its hydrophilic nature (Figure 2a). Furthermore, the transmembrane analysis of FadD18 suggests a potential lack of transmembrane characteristics, indicating its classification as an outer membrane protein and suggesting possible interactions with extracellular molecules (Figure 2b). A sequence alignment and similarity analysis demonstrate that the C-terminal regions of the FadD18 and FadD19 proteins exhibit a high degree of similarity (Figure 2c). A conserved domain analysis indicated that both the FadD18 and FadD19 proteins contain the PRK07798 superfamily domain associated with mycobacterial acyl-CoA synthetase function (Figure 2d). Additionally, FadD18 shared the conserved motif 2 with a high level of homology among mycobacteria (Figure 2d). These findings suggest that FadD18 may exhibit a comparable function to FadD19, which is known for its critical involvement in sterol metabolism in mycobacteria [15,16]. Consequently, additional research is warranted to explore the potential functions of FadD18.

### 3.3. B2909 Colonies Had a Smaller Morphology than BCG Colonies

The single-colony morphology of strain B2909 demonstrated an area approximately half that of the wild-type BCG. In regards to color uniformity, B2909 displayed a more uniform and smooth coloration compared to that of the wild-type BCG, which showed a lighter edge with more wrinkles in the center (Figure 3).

### 3.4. FadD18 Inhibited the Growth of BCG

Growth curves were used to assess the effect of FadD18 on mycobacterial extracellular survival. Overall, B2909 demonstrated a significantly higher growth rate than that of wild-type BCG between day 15 and 27 (*p* < 0.001), except for day 18 (Figure 4a). B2909 demonstrated a significantly higher intracellular survival ability than that of wild-type BCG (*p* < 0.001) 24, 48, and 72 h PI. The complementation of FadD18 (B2090C) decreased its survival ability to normal levels, indicating that FadD18 can inhibit the growth of BCG both intracellularly and extracellularly (Figure 4b).

### 3.5. FadD18 Promoted the Expression of Proinflammatory Cytokines

The mRNA and protein levels of IL-6, IL-1β, and TNF-α induced by wild-type BCG, B2909, and B2909C were assessed using RT-PCR and an ELISA. The findings suggest that B2909 resulted in significantly lower mRNA expression of IL-6, IL-1β, and TNF-α compared to that of wild-type BCG across all time points (Figure 5a–c). Conversely, at the protein level, only the expression of IL-1β and IL-6 was diminished in the B2909-infected group, with the complementation of FadD18 leading to the restoration of their expression levels to be similar to those of wild-type BCG (Figure 5d,e). Although B2909 induced lower levels of TNF-α mRNA compared to BCG and B2909C, there were no statistically significant differences in TNF-α protein levels among the groups infected with BCG, B2909, and B2909C (Figure 5f).

The findings suggest that FadD18 is capable of enhancing the expression of proinflammatory cytokines IL-1β and IL-6 at both the mRNA and protein levels, while only facilitating the expression of TNF-α mRNA.

### 3.6. FadD18 Activated the NF-κB and MAPK Signaling Pathways

In order to clarify the mechanism by which FadD18 amplifies cytokine expression, we evaluated its impact on the NF-κB and MAPK signaling pathways. B2909 or BCG were used to infect THP-1 macrophages for varying durations (0–24 h), and the phosphorylation status of key signaling molecules in these pathways was analyzed via Western blotting. The results of our study indicate that the lack of FadD18 inhibits the phosphorylation of p38, ERK, JNK, and p65 (Figure 6), implying that FadD18 plays a role in promoting the activation of BCG-induced NF-κB and MAPK signaling pathways.

## 4. Discussion

In total, 36 *fadD* genes are present in *M. tb* and are involved in lipid metabolism, the expression of FAALs associated with lipid synthesis, and the expression of FACLs associated with lipid degradation [2]. Herein, the screening of the BCG mutant library identified the *fadD* family mutant B2909 (with an inactivated *fadD18* gene), which can reduce the adhesion and invasion ability of *M. tb* to A549 cells. *fadD18* is annotated as encoding an FACL, which may be involved in lipid metabolism, but the function of the gene is unclear. To investigate the function of FadD18, the complementary strain of mutant B2909 was constructed. We found that FadD18 inhibited the BCG survival both intra- and extracellularly and promoted proinflammatory immune responses via the NF-κB and MAPK signaling pathways.

### 4.1. FadD18 Enhanced the Virulence of Mycobacteria but Inhibited Their Survival

Bacterial invasion is commonly used as a marker of virulence [17]. Our findings demonstrated that FadD18 enhances the adhesion and invasion capabilities of mycobacteria. Additionally, the morphological analysis revealed that FadD18 promotes the enlargement and wrinkling of BCG colonies, which were also associated with virulence. The essential role of FadD26 and FadD28 has been indicated as their synthesis of virulent phthiocerol dimycocerosate lipids, a finding consistent with results in our study [7]. The homology analysis of FadD18 and FadD19 indicates a close relationship between FadD18 and FadD19, which have a similar C-terminus with the same key residues (Figure 2c). This suggests that FadD18, like FadD19, is probably involved in the sterol metabolism of mycobacteria. A further analysis of conserved domains shows that both proteins possess the PRK07798 superfamily domain associated with acyl-CoA synthetase function, highlighting the importance of FadD18 in sterol and lipid metabolism [15,16]. Moreover, FadD18 was found to potentially augment the pathogenicity of the bacterium through facilitation of its adhesion to and invasion of macrophages, as well as the modulation of bacterial morphology. However, this process resulted in the inhibition of growth and survival of BCG in both extracellular and intracellular environments. As FadD18 functions as an FACL involved in lipid metabolism, the lipid synthesized by FadD18 may be linked to virulence, albeit at the expense of a significant amount of energy consumption which hampers bacterial growth and replication. This trade-off, however, may confer long-term benefits for bacterial maintenance. Nevertheless, the functional analysis of FadD18 in lipid metabolism was not conducted in this study, leaving this topic to be explored in future research.

### 4.2. FadD18 Enhanced the Production of Proinflammatory Cytokines via Activation of the NF-κB and MAPK Signaling Pathways

In the early stages of *M. tb* infection, the host’s innate immune responses, including inflammation, play a crucial role in eliminating pathogens [18]. The NF-κB and MAPK signaling pathways in macrophages play a key role in balancing pro- and anti-inflammatory cytokines during this process [19,20]. These cytokines are crucial in the host’s interaction with *M. tb*, but excessive inflammation can lead to increased bacterial invasion and damage [21]. Recent studies have elucidated the complex interplay between host signaling pathways and *M. tb* infection, highlighting the pathogen’s capacity to subvert immune responses by manipulating these pathways [22]. Furthermore, inhibiting NF-κB or MAPK pathways could result in the enhanced survival of *M. tb* within macrophages, emphasizing the essential role of these pathways in the regulation of infection [23,24].

The NF-κB transcription factor plays a critical role in modulating immune responses and inflammation by regulating the expression of immune-regulatory molecules, thereby impacting the intracellular survival of pathogens [23]. The NF-κB signaling pathway is essential for the rapid transcriptional activation of a multitude of genes involved in immune and inflammatory responses. Following infection with *M. tb*, NF-κB is activated and translocates to the nucleus, where it stimulates the expression of a range of proinflammatory cytokines, such as IL-1β, IL-6, and TNF-α [23]. These cytokines are crucial in augmenting the antimicrobial capabilities of macrophages and orchestrating the broader immune response against pathogens [25]. For example, TNF-α is crucial for the development and maintenance of granulomas, which are vital structures in containing *M. tb* infection and play a pivotal role in suppressing and eradicating *M. tb* in the host by promoting autophagy and assisting in the clearance of bacilli through autophagolysosomes [26,27,28]. IL-6 is necessary for initiating protective immune responses against mycobacteria, and the depletion of IL-6 can worsen *M. avium* infection and diminish protection against aerogenic *M. tb* infection in individuals vaccinated with culture filtrate protein [29,30]. Furthermore, IL-1β has the capacity to regulate bacterial replication via proinflammatory pathways, yet it may also exacerbate its condition by inducing excessive inflammation [31,32]. Similarly, upon infection with *M. tb*, the MAPK signaling pathway is triggered, leading to the modulation of multiple cellular functions, such as cytokine synthesis, cellular differentiation, and programmed cell death [33,34,35]. The activation of MAPK pathways, specifically p38, ERK, and JNK, occurs in infected macrophages, resulting in the generation of proinflammatory cytokines and chemokines that facilitate the recruitment and activation of additional immune cells [35]. Research has shown that the p38 MAPK pathway plays a crucial role in promoting the synthesis of IL-12, a cytokine necessary for the differentiation of Th1 cells and the subsequent production of IFN-γ [36]. This cytokine, in turn, stimulates macrophages to eradicate intracellular *M. tb*.

*M. tb* has developed various virulent effector proteins that specifically target key components of the innate immune system, disrupting host defenses. *M. tb* YrbE3A has been shown to induce cytokine production by phosphorylating p65, p-JNK, and p38, thereby activating the NF-κB and MAPK pathways [37]. Additionally, the *M. tb* protein Rv2346c has been found to enhance mycobacterial survival by inhibiting the production of TNF-α and IL-6 through the p38/miRNA/NF-κB pathway in macrophages [38]. Furthermore, Rv0309 and PtpA have been demonstrated to reduce TNF-α and IL-1β expression, leading to an increase in the bacterial burden in the lungs of BCG-challenged mice [39,40]. The regulation of proinflammatory cytokine secretion by mycobacterium *fadD* family genes, specifically *fadD13* and *fadD33*, involves the modulation of the NF-κB signaling pathway and the MAPK signaling pathway, respectively [8,41]. In our study, we found that the expression of the *fadD18* gene significantly increased the secretion of IL-1β, IL-6, and TNF-α through the activation of the NF-κB and MAPK signaling pathways. To investigate the potential involvement of the NF-κB and MAPK signaling pathways in the effect of FadD18 on cytokine production, we assessed the phosphorylation levels of p65, p38, JNK, and ERK1/2 in THP-1 macrophages. These findings suggest that FadD18 influences the phosphorylation levels of proteins within a specific pathway. However, the specific receptors responsible for recognizing mutant strains and triggering downstream pathway signaling remain unclear. We hypothesize that FadD18 may interact with macrophage surface receptors to activate the NF-κB and MAPK signaling pathways and induce the release of proinflammatory cytokines. Additionally, the inflammatory response of the mutant strain is suppressed upon the infection of macrophages, ultimately enhancing its intracellular survival. Therefore, the promotion of cytokine production by FadD18 is probably responsible for inhibiting mycobacterial survival.

## 5. Conclusions

Our study demonstrates that FadD18 plays a role in enhancing the adhesion and invasion abilities of mycobacteria in human lung epithelial cells. Additionally, FadD18 suppresses BCG intracellular survival by stimulating the production of proinflammatory cytokines via the NF-κB and MAPK signaling cascades in macrophages (Figure 7). This research highlights a novel mycobacterial protein that influences the adhesion, invasion, inflammation, and intracellular survival mechanisms employed by *M. tb* to overcome cellular barriers for its transmission, indicating the protein’s potential as a target for therapeutic interventions and in vaccine development for tuberculosis.

## Figures and Tables

**Figure 1 cells-13-01019-f001:**
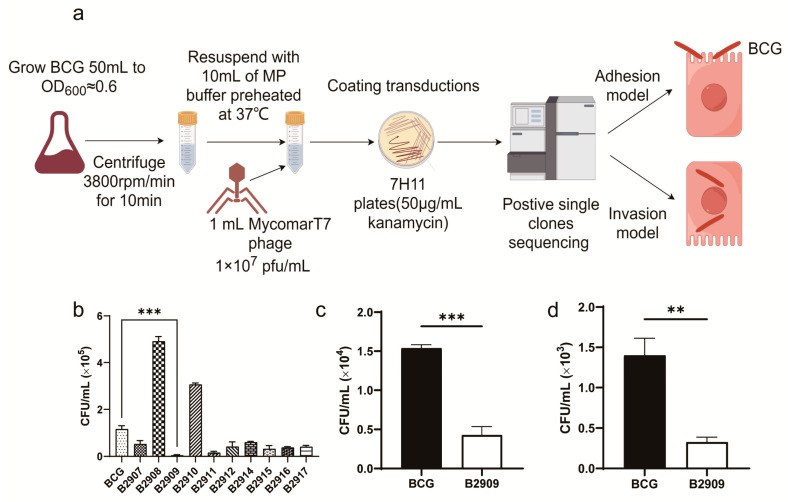
Evaluation of invasion and adhesion ability of BCG. An MOI of 10:1 was used for infecting A549 cells with BCG strains at 37 °C for 1 h for the invasion assay or 4 °C for 30 min for the adhesion assay. The CFU assay was used to quantify the number of intracellular BCG mutants. (**a**) The workflow for BCG transposon mutant library construction and screening. (**b**) A comparative analysis was undertaken to evaluate the invasive potential of different BCG mutants, with each mutant strain being assessed against the wild-type BCG in all tests. (**c**) Invasion and (**d**) adhesion of A549 cells by both the wild-type (BCG) and B2909 mutants. Data are presented as means ± SEMs, with significance denoted by ** for *p* < 0.01 and *** for *p* < 0.001. Each sample was assessed in triplicate.

**Figure 2 cells-13-01019-f002:**
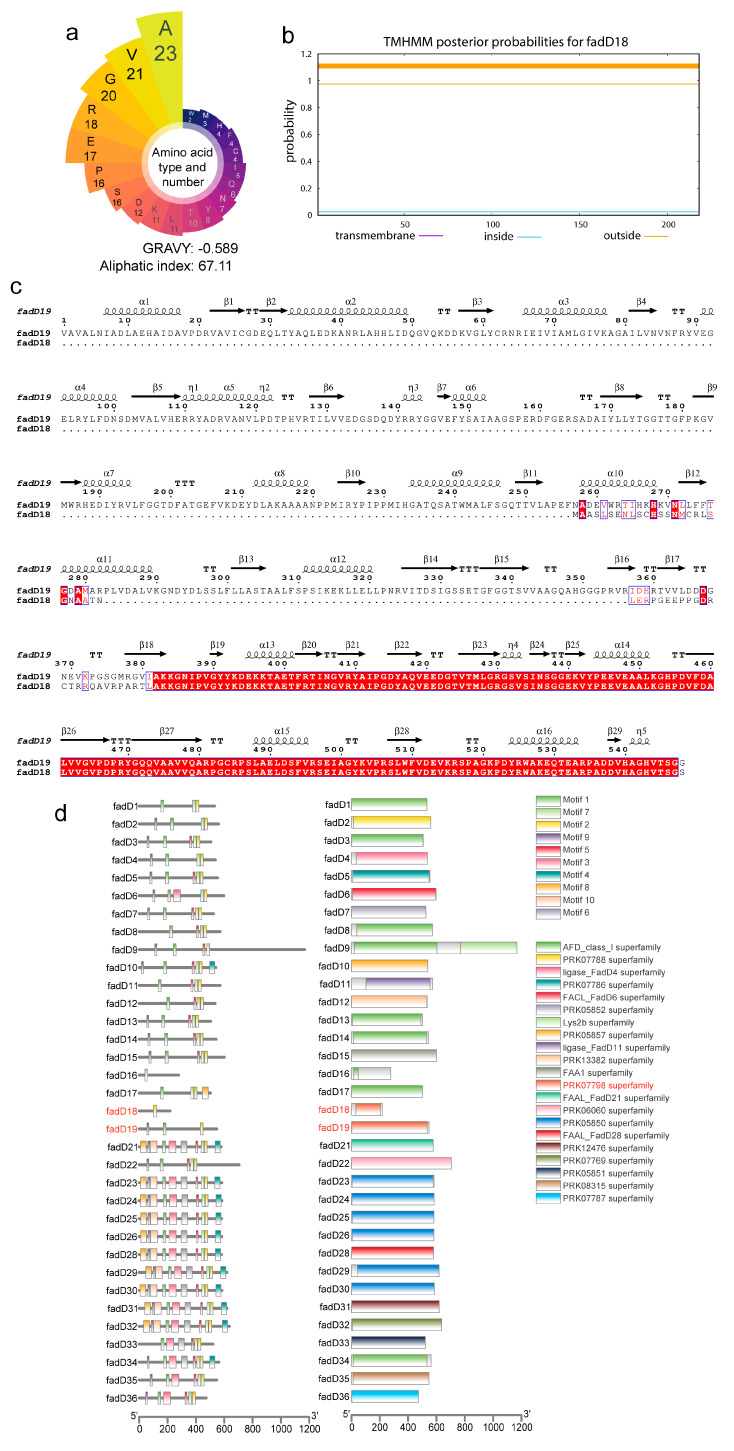
Bioinformatic analysis of FadD18. (**a**) Type and number of amino acids of FadD18. The ProtParam web server was used to determine the grand average of hydropathicity index (GRAVY) values and the aliphatic index of the FadD18. (**b**) TMHHM2.0 was used to predict transmembrane domains. (**c**) Sequence alignment of FadD18 and FadD19. An alignment of protein homologs was constructed with Clustal Omega and ESPript 3.0. Red represents identity. (**d**) Domains and conserved motifs identified in FadD18 and homologs. The conserved domain in FadD18 and its homologs was analyzed using the NCBI CDD search tool, while the MEME suite motif search tool was utilized to identify conserved motifs in these protein sequences.

**Figure 3 cells-13-01019-f003:**
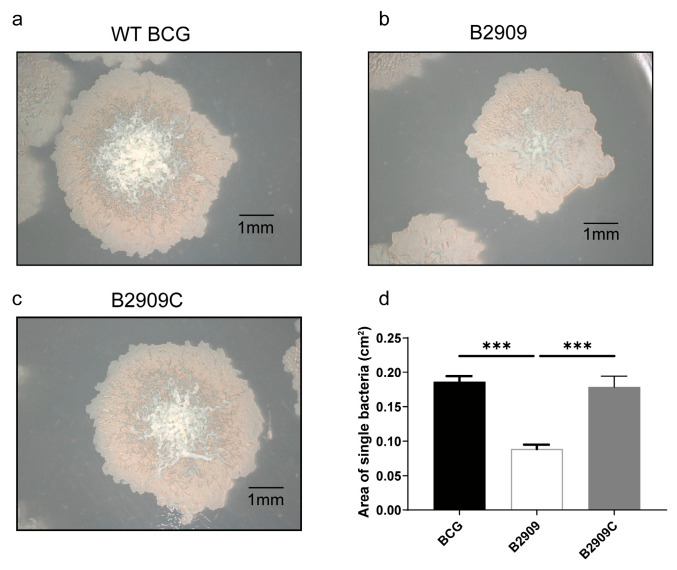
Growth rate and morphology of B2909 strain. Morphology and area of single colonies of wild-type BCG (BCG) (**a**), B2909 (**b**), and B2909 complement (B2909C) (**c**) BCG strains. (**d**) Area of colonies of BCG, B2909, and B2909C BCG strains. *** *p* < 0.001.

**Figure 4 cells-13-01019-f004:**
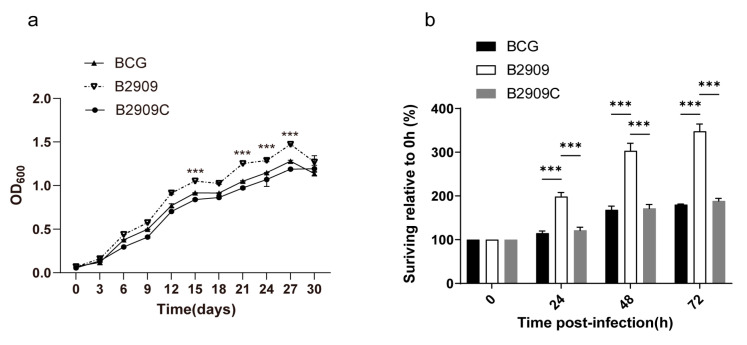
Growth curve and survival ability of BCG strains. (**a**) The growth curves of the BCG, B2909, and B2909C strains were evaluated in 7H9 liquid medium at 37 °C by monitoring the optical density at 600 nm every three days. (**b**) THP-1 cells were infected with BCG, B2909, and B2909C strains at an MOI of 10:1 for 4 h. Following infection, cells were washed with PBS three times and then maintained in fresh media. Cell lysis was carried out 0, 24, 48, or 75 h PI, and CFUs of intracellular bacteria were quantified by plating cell lysates on 7H11 agar plates, followed by a 3-week incubation period for colony enumeration. Data are expressed as means ± standard errors of the means. Significance was determined as *** *p* < 0.001.

**Figure 5 cells-13-01019-f005:**
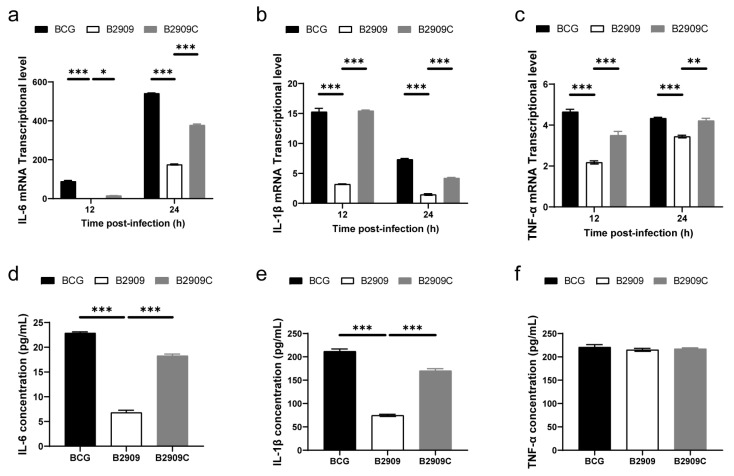
Expression of proinflammatory cytokines in THP-1 macrophages. Subsequently, macrophages were harvested at 12 and 24 h. The mRNA expression levels of IL-6 (**a**), IL-1β (**b**), and TNF-α (**c**) were quantified by RT-qPCR. The protein concentrations of IL-6 (**d**), IL-1β (**e**), and TNF-α (**f**) were measured using ELISA. The mRNA levels of cytokines were standardized relative to the levels of β-actin mRNA. Data are presented as means ± SEMs. Significance is indicated as * *p* < 0.05, ** *p* < 0.01, *** *p* < 0.001.

**Figure 6 cells-13-01019-f006:**
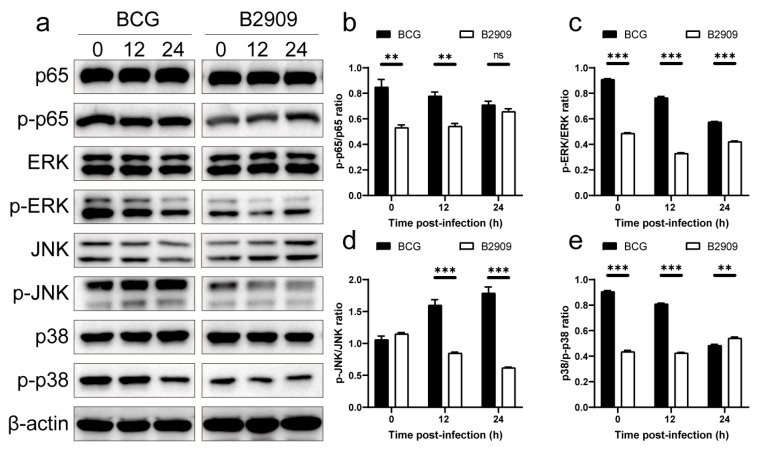
FadD18 significantly augmented the activation of NF-κB and MAPK signaling pathways induced by BCG. (**a**) THP-1 macrophages were treated with BCG and mutant strain B2909 at an MOI of 10:1 for a specific infection time. Cells were then lysed on ice for 30 min and analyzed via Western blotting to measure the levels of total and phosphorylated forms of p65, p38, JNK, and ERK1/2 proteins. (**b**–**e**) The gray-scale values of p-p65/p65 (**b**), p-p38/p38 (**c**), p-JNK/JNK (**d**), and p-ERK/ERK (**e**) were determined. Data are presented as means ± SEMs. Significance is indicated as ns *p* > 0.05, ** *p* < 0.01, *** *p* < 0.001.

**Figure 7 cells-13-01019-f007:**
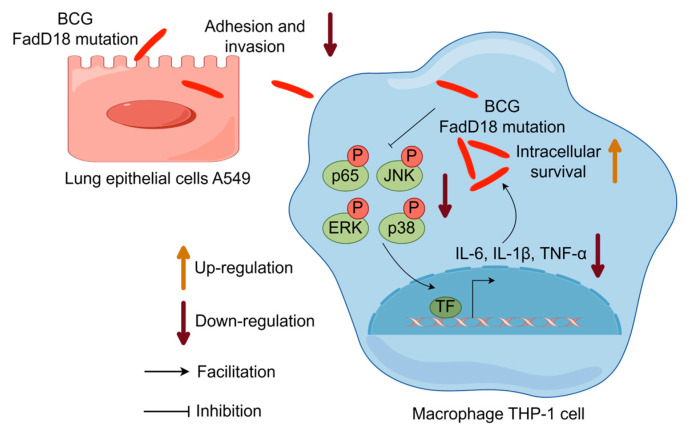
Proposed model of FadD18 interaction with host cells. Schematic of the proposed interaction of FadD18 with host cells, showing the enhancement of mycobacteria’s adhesion and invasion capabilities, as well as the promotion of proinflammatory cytokine production through the NF-κB and MAPK signaling pathways.

**Table 1 cells-13-01019-t001:** Primers Used in This Study.

Primer Name	Sequence (5′-3′)	Products (bp)
IL-6	F: ACTCACCTCTTCAGAACGAA	149
	R: CCATCTTTGGAAGGTTCAGG	
IL-1β	F: GTGGCAATGAGGATGACTTGTTC	120
	R: GGTGGTCGGAGATTCGTAGCT	
TNF-α	F: GGAGAAGGGTGACCGACTCA	70
	R: CTGCCCAGACTCGGCAA	
β-actin	F: CATGTACGTTGCTATCCAGGC	250
	R: CTCCTTAATGTCACGCACGAT	

## Data Availability

The datasets generated in this article and the Appendix A are available upon request from the corresponding authors.

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
