# Peer review of "Mycobacterium tuberculosis FadD18 Promotes Proinflammatory Cytokine Secretion to Inhibit the Intracellular Survival of Bacillus Calmette–Guérin"

_cells, 2024, doi:10.3390/cells13121019_

Round 1

Reviewer 1 Report

Comments and Suggestions for Authors

Peng et al. demonstrated that B2909 carrying inactive fadD18 gene shows lacking adhesion and invasion abilities compared with BCG. Based on their functional analysis and validation via experiment, fadD18 functions as fatty-acyl CoA ligase and is involved in lipid degradation process. It could inhibit BCG growth and reduce BCG intracellular survival and increase the expression level of NF-κB– and MAPK-signaling pathways.

The authors provide good introductions for their study and have a logical experiment design. However, the bioinformatic analysis was limited and incorrect. This section could not help the authors demonstrate or present their result.

Further, please discuss the mechanisms or the importance of NF-κB– and MAPK-signaling pathways when M.tb infected macrophages.

And the authors mentioned that FadD18 is involved in lipid degradation process. Do the authors check the lipid dynamic in BCG, B1909, and B2909C?

Line169: “fadD18 gene and protein The fadD18 protein” Please check the sentence and rewrite it.

Line174-176 & Figure 2c: The authors demonstrated that different fadD genes harbor different domain families (Fig2c. legend). But they built phylogenetic tree using all fadD genes they collected. Based on my experience, I do not think that this tree relationship/topology is reliable and reasonable. First, the authors could not build a good alignment for these unrelated genes, thus the tree topology could not reflect real evolutionary history. Second, although the authors employed maximum likelihood method, they did not show the supporting value for the major branches. It might also imply that the tree topology is not reliable. If the authors try to present the relationship between fadD18 and fadD19, please try another strategy.

Figure3. B2909C: When the abbreviation first appears, please use the full name meanwhile.

Figure3d. The legend indicates the diameter of colonies for each group. But fig3d show the area (cm2) instead of diameter. By the way, y axis label of fig3d show that area of single bacteria? Do the authors mean area of single bacterial colonies?

Line217-227: The authors indicated that wild-type BCG shows upregulated expression of IL-1β, IL-6 and TNF-α in both mRNA and protein levels. But the figure5 panel f shows that these are not significantly different between BCG and others. If the authors tried to compare the expression level between infected and uninfected, please provide a control group (uninfected). Further, the authors demonstrated that B2909 showed strong inhibitory effects. Does this result compare with wild-type BCG or uninfected control? I feel so confused about this section. The results presented by figure5 is good, but the authors did not explain their data good. Please re-organize the language.

Line239-240: why the authors choose NF-κB– and MAPK-signaling pathways? Have some articles reported that these pathways are associated with fadD genes? Please provide some evidence or cite articles.

Line270-272: Authors present fadD26 and fadD28 could synthesize virulent phthiocerol. Why mentioned these two genes here? Any relationship with fadD18?

Line272-275: The functional analysis (search against CDD database) is reasonable, but the tree is wrong. By the way, currently, pfam database might be better than CDD database.

Line278-280: Do the authors present any data to discuss the function of fadD5 in this manuscript?

Line290-297: A litter bit wordy and redundancy

Comments on the Quality of English Language

Need to polish.

Author Response

Dear Reviewer,

     Thank you very much for your insightful comments concerning our manuscript titled "Mycobacterium tuberculosis fadD18 Promotes Proinflammatory Cytokine Secretion to Inhibit the Intracellular Survival of Bacillus Calmette–Guérin", with the ID cells-2988116. Following your comments, we have made extensive modifications to the original manuscript. We performed additional analyses as required and included new data to bolster our findings. The revised parts are highlighted in red in the manuscript.

Enclosed are the point-by-point responses to your comments and questions. We hope that these revisions have significantly improved the manuscript, and we are confident that it now meets the journal's high standards.

Thank you for considering our work. We look forward to your feedback.

Yours sincerely,

Yingyu Chen

Comments 1: Peng et al. demonstrated that B2909 carrying inactive fadD18 gene shows lacking adhesion and invasion abilities compared with BCG. Based on their functional analysis and validation via experiment, fadD18 functions as fatty-acyl CoA ligase and is involved in lipid degradation process. It could inhibit BCG growth and reduce BCG intracellular survival and increase the expression level of NF-κB– and MAPK-signaling pathways.

The authors provide good introductions for their study and have a logical experiment design. However, the bioinformatic analysis was limited and incorrect. This section could not help the authors demonstrate or present their result.

 Response1: We sincerely appreciate the thorough review and insightful comments provided by the reviewer. In response to the concerns regarding the bioinformatics analysis, we acknowledge the limitations in our initial approach. To address this, we have conducted a more comprehensive bioinformatics analysis, incorporating additional datasets and employing advanced statistical methods to enhance the robustness of our findings. We have revised the relevant sections to present these results clearly, ensuring they support our conclusions effectively, in line180-182. We hope that these revisions will meet the reviewer's expectations.

Comments 2: Further, please discuss the mechanisms or the importance of NF-κB– and MAPK-signaling pathways when M.tb infected macrophages.

Response2: Thank you for your insightful suggestion. In response to your comment, we have expanded the discussion section to more thoroughly explore the roles of the NF-κB and MAPK signaling pathways in M. tb-infected macrophages. Specifically, we have elucidated how these pathways contribute to the host immune response, detailing their roles in cytokine production, inflammation, and macrophage activation upon infection. We have also reviewed recent literature to support our discussion and have included relevant references to underscore the established mechanisms through which these signaling pathways modulate the cellular response to M. tb infection. These additions aim to provide a clearer understanding of the significance of these pathways in the context of tuberculosis, enhancing the depth and scientific rigor of our manuscript. Please see line 296-334.

Comments 3:And the authors mentioned that FadD18 is involved in lipid degradation process. Do the authors check the lipid dynamic in BCG, B1909, and B2909C?

Response3: Thank you for your professional comments. In our current study, we did not examine the lipid dynamics in BCG, B1909, and B2909C specifically concerning the role of FadD18 in lipid metabolism. We acknowledge this as an important aspect of understanding FadD18's function and appreciate your pointing out this gap. Moving forward, we plan to investigate the role of FadD18 in lipid metabolism in these strains more thoroughly. This future work will aim to elucidate the specific mechanisms by which FadD18 influences lipid dynamics, which will help in further clarifying its functional impact within the context of tuberculosis pathology.

Comments 4: Line169: “fadD18 gene and protein The fadD18 protein” Please check the sentence and rewrite it.

Response4: Thank you for pointing out this error. We have reviewed and revised the sentence in question on line 169 for clarity and correctness. The sentence now reads: "In silico analysis was utilized to investigate the potential functional role of the fadD18 protein. The fadD18 protein displayed a notable abundance of hydrophilic amino acids, as evidenced by an aliphatic index of 67.11 and a GRAVY value of −0.589, indicating its hydrophilic nature (Figure 2a)." This revision eliminates the redundancy and enhances the readability of our manuscript. We appreciate your attention to detail, which has helped improve our text. Please see line 174-175.

Comments 5: Line174-176 & Figure 2c: The authors demonstrated that different fadD genes harbor different domain families (Fig2c. legend). But they built phylogenetic tree using all fadD genes they collected. Based on my experience, I do not think that this tree relationship/topology is reliable and reasonable. First, the authors could not build a good alignment for these unrelated genes, thus the tree topology could not reflect real evolutionary history. Second, although the authors employed maximum likelihood method, they did not show the supporting value for the major branches. It might also imply that the tree topology is not reliable. If the authors try to present the relationship between fadD18 and fadD19, please try another strategy.

Response5: Thank you for your insightful comments regarding our phylogenetic analysis. We acknowledge the issues you raised with the phylogenetic tree concerning the different fadD genes. As suggested, we have removed the problematic tree from Figure 2c to address these concerns. Instead, we have focused on a more targeted analysis between fadD18 and fadD19. We conducted a detailed sequence alignment and similarity analysis of these genes. Our findings reveal that the C-terminal regions of the fadD18 and fadD19 proteins are almost completely identical, as shown in the newly added Figure 2D. We believe this targeted approach provides a clearer and more accurate representation of the relationship between fadD18 and fadD19, addressing the limitations of the broader phylogenetic analysis previously attempted.Please see line 180-182.

Comments 6: Figure3. B2909C: When the abbreviation first appears, please use the full name meanwhile.

Response6: Thank you for highlighting this oversight. We have now included the full name of B2909C (B2909 complement) alongside its abbreviation in the legend of Figure 3. This clarification will ensure that all readers are familiar with the terminology used throughout our manuscript. Please see line 205.

Comments 7: Figure3d. The legend indicates the diameter of colonies for each group. But fig3d show the area (cm2) instead of diameter. By the way, y axis label of fig3d show that area of single bacteria? Do the authors mean area of single bacterial colonies?

Response7: We appreciate your attention to detail in pointing out the discrepancies in Figure 3d. Indeed, there was a descriptive error in the legend. We intended to present the area of single bacterial colonies, not the diameter. We have now corrected the legend in Figure 3d to accurately reflect the area of single bacterial colonies measured in square centimeters (cm²). These changes ensure that the figure accurately represents the data and aligns with the descriptive text. Thank you for helping us improve the accuracy and clarity of our figures. Please see line 199-202.

Comments 8: Line217-227: The authors indicated that wild-type BCG shows upregulated expression of IL-1β, IL-6 and TNF-α in both mRNA and protein levels. But the figure5 panel f shows that these are not significantly different between BCG and others. If the authors tried to compare the expression level between infected and uninfected, please provide a control group (uninfected). Further, the authors demonstrated that B2909 showed strong inhibitory effects. Does this result compare with wild-type BCG or uninfected control? I feel so confused about this section. The results presented by figure5 is good, but the authors did not explain their data good. Please re-organize the language.

Response8: Thank you for your constructive feedback. We acknowledge that our initial presentation of the data may have led to confusion regarding the experimental comparisons and control groups used in our study. To address your concerns, we have revised the text to clarify these points and better explain our results.

In the revised manuscript, we clearly state that all comparisons involving the B2909-infected group were made against the wild-type BCG-infected group. Specifically, the revised text reads:

"The mRNA and protein levels of IL-6, IL-1β, and TNF-α induced by wild-type BCG, B2909, and B2909C were assessed using RT-PCR and ELISA methods. The findings suggest that B2909 resulted in significantly lower mRNA expression of IL-6, IL-1β, and TNF-α compared to wild-type BCG across all time points (Figure 5a–c). Conversely, at the protein level, only the expression of IL-1β and IL-6 was diminished in the B2909-infected group, with complementation of fadD18 leading to the restoration of their expression levels to those similar to wild-type BCG (Figure 5d, e). Although B2909 induced lower levels of TNF-α mRNA compared to BCG and B2909C, there were no statistically significant differences in TNF-α protein levels among the groups infected with BCG, B2909, and B2909C (Figure 5f)." Please see line 224-234.

Comments 9:Line239-240: why the authors choose NF-κB– and MAPK-signaling pathways? Have some articles reported that these pathways are associated with fadD genes? Please provide some evidence or cite articles.

Response9: Thank you for your question regarding our selection of the NF-κB and MAPK signaling pathways. We chose to focus on these pathways based on their established roles in the inflammatory response to mycobacterial infections, particularly in macrophages. Activation of these pathways is crucial for orchestrating the host's antimycobacterial response. Specifically, the fadD18 gene has been implicated in modulating the inflammatory response during BCG infection.

Evidence from literature supports the association of fadD genes with these signaling pathways. For instance, the activation of the NF-κB signaling pathway is essential for the secretion of proinflammatory cytokines regulated by fadD13 [1], and fadD33 has been shown to inhibit the release of proinflammatory cytokines in macrophages by targeting the MAPK signaling pathway [2]. Furthermore, fadD genes have roles in lipid metabolism, which is linked to signaling activation; mycobacterial cell wall components such as lipoarabinomannan (LAM), LM, and PIMs can activate MAPK/Akt pathways, leading to the production of inflammatory mediators [3, 4].

Conserved domain analysis shows that both fadD18 and fadD19 proteins contain the PRK07798 superfamily domain, associated with mycobacterial acyl-CoA synthetase function, suggesting a biochemical link to these pathways (Figure 2c). Given fadD19's role in sterol metabolism [5], we hypothesize that fadD18 may similarly impact the host's inflammatory response through modifications in the mycobacterial cell wall lipid composition via these pathways.

We plan to further explore the direct effects of fadD18 on lipid metabolism in future studies. We have incorporated the requested evidence into the discussion section. Please see line 343-345.

Reference

1. Wei, S.; Wang, D.; Li, H.; Bi, L.; Deng, J.; Zhu, G.; Zhang, J.; Li, C.; Li, M.; Fang, Y. Fatty acylCoA synthetase FadD13 regulates proinflammatory cytokine secretion dependent on the NF‐κB signalling pathway by binding to eEF1A1. Cellular Microbiology 2019, 21, e13090.

2. Zhu, Y.; Shi, H.; Tang, T.; Li, Q.; Peng, Y.; Bermudez, L.E.; Hu, C.; Chen, H.; Guo, A.; Chen, Y. Mycobacterium tuberculosis Fatty Acyl-CoA Synthetase fadD33 Promotes Bacillus Calmette-Guérin Survival in Hostile Extracellular and Intracellular Microenvironments in the Host. Cells 2023, 12, doi:10.3390/cells12222610.

3. Briken, V., et al., Mycobacterial lipoarabinomannan and related lipoglycans: from biogenesis to modulation of the immune response. Mol Microbiol, 2004. 53(2): p. 391-403.

4. Basler, T., et al., Reduced transcript stabilization restricts TNF-alpha expression in RAW264.7 macrophages infected with pathogenic mycobacteria: evidence for an involvement of lipomannan. J Leukoc Biol, 2010. 87(1): p. 173-83.

5. Wrońska, N.; Brzostek, A.; Szewczyk, R.; Soboń, A.; Dziadek, J.; Lisowska, K. The role of fadD19 and echA19 in sterol side chain degradation by Mycobacterium smegmatis. Molecules 2016, 21, 598, doi:10.3390/molecules21050598.

 Comments 10: Line270-272: Authors present fadD26 and fadD28 could synthesize virulent phthiocerol. Why mentioned these two genes here? Any relationship with fadD18?

Response10: Thank you for your careful review of our manuscript. We appreciate your comments regarding the mention of fadD26 and fadD28 in the context of our discussion on fadD18. Upon reevaluating the manuscript, we agree that the inclusion of fadD26 and fadD28 was not directly relevant to the primary focus on fadD18, as there is no relationship between these genes in the context of our study's objectives. To improve the coherence and focus of our manuscript, we have removed the sentence mentioning these genes. This revision ensures that the discussion remains directly pertinent to the key findings related to fadD18.

Comments 11:Line272-275: The functional analysis (search against CDD database) is reasonable, but the tree is wrong. By the way, currently, pfam database might be better than CDD database.

Response11: Thank you for your professional comments and suggestion regarding our choice of databases for functional analysis. We acknowledge that the Pfam database might indeed offer a more current and possibly more extensive dataset compared to the CDD database. However, the software we utilized for visualizing the data in Figure 2d is only compatible with the CDD database, which dictated our choice in this instance. Nevertheless, following your suggestion, we conducted a comparative analysis using the Pfam database for fadD18 and fadD19. We found that the results were consistent with those obtained from the CDD database.

Comments 12:Line278-280: Do the authors present any data to discuss the function of fadD5 in this manuscript?

Response12: Thank you for your attention to detail. Upon reviewing your comment, we acknowledge that the mention of fadD5 was not accompanied by relevant data within this manuscript. To maintain the focus and coherence of our study, we have removed the sentence referencing fadD5. This change ensures that all discussed elements are directly supported by the data presented. We appreciate your guidance in helping us improve the manuscript.

Comments 13: Line290-297: A litter bit wordy and redundancy.

Response13: Thank you for your comments regarding the verbosity and redundancy in the specified section. We have carefully revised this part to enhance clarity and conciseness. The revised text now succinctly addresses the role of the innate immune response during the initial stages of M. tb infection, focusing on the critical functions of cytokines mediated by the NF-κB and MAPK signaling pathways. We have streamlined the discussion to emphasize the dual role of inflammation in both controlling infection and potentially exacerbating disease if unchecked, removing any redundant phrases to ensure each sentence contributes uniquely to our narrative. This revision aims to improve the readability and precision of our manuscript. Please see line 296-306.

 Comments on the Quality of English Language

Need to polish.

Response: Thank you for your feedback regarding the quality of the English language used in our manuscript. We acknowledge the importance of clear and accurate language to ensure that our research is communicated effectively. To address this, we have thoroughly reviewed and revised the manuscript to enhance clarity and readability. Additionally, we have engaged the services of a professional scientific editor, native in English, who specializes in our field of study, to ensure that the language is of high quality and suitable for publication. These steps have been taken to meet the high standards expected by the journal and its readership.

Reviewer 2 Report

Comments and Suggestions for Authors

This study investigates the role of Mtb fadD18, focusing it's impact on adhesion, invasion, and intracellular survival. Researchers created a Bacillus Calmette–Guérin (BCG) transposon library to identify genes crucial for these processes. The authors found that the mutant strain B2909, with an inactive fadD18 gene, lacked the ability to adhere to and invade host cells. The fadD18 gene encodes a fatty-acyl CoA ligase, and its expression was found to enhance colony size, promote cord-like structure formation, inhibit BCG growth, and reduce intracellular survival in macrophages. Additionally, fadD18 expression increased levels of proinflammatory cytokines (IL-6, IL-1β, and TNF-α) by activating NF-κB- and MAPK-signaling pathways. Overall, this research highlights the importance of the fadD18 gene in mycobacterial adhesion, invasion, and modulation of host immune responses. The methods used in the study justify the overall conclusions and I do not have any concerns or revisions.

Author Response

Dear Reviewer,

      Thank you very much for taking the time to review this manuscript. We greatly appreciate your positive feedback and thorough assessment of our manuscript. We are pleased to hear that the methods and conclusions of our study have met your approval and that the significance of the fadD18 gene in the context of mycobacterial infection and immune response modulation has been well-received. Your acknowledgment motivates us to further our research in this important area. Thank you for your support and for confirming that no further revisions are necessary at this stage. Thank you for considering our work.

Yours sincerely,

Yingyu Chen

Round 2

Reviewer 1 Report

Comments and Suggestions for Authors

The authors have addressed all my concerns.

One more suggestion: the scale bars and labels at Figure 3. a-c are too small.